# Outer membrane vesicles from *Aeromonas veronii*: biological properties and synergistic immunity enhancement with inactivated bacteria in crucian carp

Zong-Xiu Wu,[1,2] Ding-Jie An,[3,4] Yi-Xuan Kou,[4] Qing Yang,[1,2] Zhi-Qiang Zhang,[5] Li Chen,[5] Guang-Ping Gao,[5] Gui-Sheng Gao,[5] Bin-Tong Yang,[1,2,4] Yuan-Huan Kang[1,2,4]

**ABSTRACT** *Aeromonas veronii* (*A. veronii*) causes hemorrhagic septicemia and enteritis in aquatic animals, posing a threat to global aquaculture. Conventional vaccines are inadequate due to short immune duration, narrow antigenic spectrum, and insufficient commercialization. Outer membrane vesicles (OMVs), naturally secreted by Gram-negative bacteria, hold significant potential in vaccine development due to their enrichment with immunologically active components. This study aimed to characterize the biological properties of OMVs derived from the *A. veronii* TH0426 strain and evaluate their potential as a vaccine. Characterization revealed a diameter distribution of 10 to 300 nm. Proteomic analysis identified 76 proteins, including conserved antigens such as outer membrane channel-forming protein II (OmpII; 80% coverage) and outer membrane protein A (OmpA; 44% coverage). Evaluation in a crucian carp (*Carassius auratus*) model demonstrated that OMVs alone or combined with inactivated whole vaccine (Av) cells significantly enhanced serum antibody titers, serum bactericidal activity, and related immune enzymes ($P < 0.05$). Furthermore, they upregulated the expression of immune factors (TGF-β, TNF-α, IL-10, IL-1β) in tissues such as the liver and spleen ($P < 0.05$). In the challenge test, the relative percent survival of the OMV + Av group reached 66%. These results indicate that *A. veronii* OMVs possess application potential as vaccines, can effectively enhance the immune protection efficacy of aquatic animals, and provide novel insights for the development of next-generation vaccines against *A. veronii*.

**IMPORTANCE** This study demonstrated the biological characteristics and application potential of OMVs derived from the *A. veronii* TH0426 strain as a vaccine. OMVs exhibited favorable safety profiles in crucian carp and significantly enhanced serum bactericidal activity. Furthermore, OMVs displayed potent immunogenicity, effectively elevating the levels of key serum immune enzymes, immune factors, and IgM, thereby significantly boosting both the innate and adaptive immune responses in crucian carp. Critically, OMVs effectively enhanced the immunoprotective efficacy conferred by inactivated *A. veronii* whole cells, significantly improving the resistance of crucian carp to *A. veronii* infection. Future studies will explore the broader application value of OMVs and further investigate the feasibility of utilizing OMVs from the *A. veronii* TH0426 strain as a potential vaccine.

**KEYWORDS** *Aeromonas veronii*, outer membrane vesicles, immunogenicity, immune protection

Aeromonas veronii (*A. veronii*) is a critical pathogen in the aquaculture industry, capable of causing severe diseases in fish, including hemorrhagic septicemia and enteritis, which pose a persistent threat to global aquaculture (1, 2). To date,

**Peer Reviewers** Wenbin Wang, Jiangsu Ocean University, Lianyungang, Jiangsu, China; Chentao Lin, Fujian Normal University College of Life Science, Fuzhou, Fujian, China; Erdal Özbek, Dicle Universitesi Tip Fakultesi, Diyarbakir, Bağlar, Türkiye

Address correspondence to Bin-Tong Yang, yangbintong@sdu.edu.cn, or Yuan-Huan Kang, kangyuanhuan@sdu.edu.cn.

Zong-Xiu Wu and Ding-Jie An contributed equally to this article. The author order was determined by increasing seniority.

The authors declare no conflict of interest.

See the funding table on p. 14.

the prevention and control of *A. veronii*-associated diseases have primarily relied on antibiotics. However, the widespread use of antibiotics has led to increasingly prominent issues, including enhanced antibiotic resistance, compromised host immunity, drug residues, and environmental pollution (3, 4). Consequently, the development of safe and effective alternative control strategies is imperative. Vaccines are widely regarded as one of the most effective means of preventing bacterial diseases in fish, owing to their high specificity, minimal side effects, and long-lasting immune protection (5). Although research on vaccines against *A. veronii* (including inactivated vaccines, live attenuated vaccines, and subunit vaccines) has made progress, commercial vaccines remain in the development or clinical trial stages, and their safety and immunogenic efficacy require further comprehensive evaluation (6–8). Therefore, developing novel *A. veronii* vaccines that are both safe and highly efficacious remains a key and urgent challenge in the field of aquatic disease control.

Outer membrane vesicles (OMVs), which are naturally secreted nano-scale membrane structures from Gram-negative bacteria, exhibit potent immunostimulatory properties and hold significant potential as vaccines and efficient delivery systems for exogenous antigens (9–12). The lipid bilayer membrane of OMVs can encapsulate various conserved antigens (e.g., outer membrane proteins, virulence-associated factors, and other immunological targets), forming multi-epitope antigen complexes capable of eliciting specific immune responses. Simultaneously, molecules on the vesicle surface, such as lipopolysaccharide (LPS) and lipoproteins, can activate dendritic cells and macrophages through pattern recognition receptors like Toll-like receptor 4 (TLR4) and Toll-like receptor 2 (TLR2), inducing robust humoral and cellular immune responses. This intrinsic property of carrying both antigens and immune-stimulating molecules confers vaccine functionality upon OMVs (13–17). The successful application of the group B meningococcal OMV vaccine in 35 countries, along with breakthroughs in preclinical research on OMV vaccines for pathogens such as *Haemophilus* influenzae and *Helicobacter pylori*, collectively underscores the significant potential of OMVs as alternatives to traditional vaccines (18–21). Despite the excellent performance demonstrated by OMVs in vaccines against various pathogens, research on *A. veronii* OMVs remains in its nascent stages.

Therefore, this study focuses on the *A. veronii* TH0426 strain. First, OMVs were extracted and purified, and their morphological characteristics and proteomic composition were analyzed to identify potential antigenic components. Second, crucian carp (*Carassius auratus*) was used as a model, and the study systematically evaluated the specific immune responses (including serum antibody titers and serum bactericidal activity) and non-specific immune reactions induced by OMVs. Finally, this study sought to elucidate the immune response to OMVs by analyzing the differential expression of pertinent immune factors in various tissues of crucian carp, including the liver, spleen, kidney, gut, and gill. This research assessed the potential of *A. veronii* OMVs as a vaccine, providing a theoretical foundation for the development of novel vaccines. This research assessed the potential of OMVs as a vaccine, providing a theoretical foundation for the development of novel vaccines.

## RESULTS

### Analysis results of the OMV components

The SDS-PAGE identification results revealed that the proteins contained in the OMVs of *A. veronii* TH0426 were mainly concentrated in the range of 40–50 kDa (see Fig. S1A at https://doi.org/10.6084/m9.figshare.30870761). After agarose gel electrophoresis, the DNA fragments ranged from 1,000 bp to 2,000 bp, micro-spectrophotometric analysis yielded a DNA concentration of 25.34 ng/µL (see Fig. S1B at https://doi.org/10.6084/m9.figshare.30870761). The endotoxin concentration in the OMV suspension was determined using an endotoxin ELISA kit, and the resulting equation was $y = -75.95x^2 + 203.14x - 12.789$, with $R^2 = 0.9961$. The endotoxin concentration was 0.294 ng/mL (2.94 EU/mL) (see Fig. S1C at https://doi.org/10.6084/m9.figshare.30870761). The protein standard curve equation was $y = 0.0507x + 0.1842$, with $R^2 = 0.997$, and the

protein concentration in the OMV suspension was 2.037 mg/mL (see Fig. S1D at https://doi.org/10.6084/m9.figshare.30870761).

## Morphology of OMVs

Under transmission electron microscopy, most of the OMVs were spherical vesicle structures, and some were irregular in shape. Clear membrane-like structures were visible, and the size of the OMVs varied, with diameters ranging from approximately 10 to 300 nm (Fig. 1B).

## Mass spectrometry analysis results

The protein composition of OMVs derived from *A. veronii* TH0426 was analyzed via LC-MS/MS. A total of 76 proteins were identified (see Table S2 at https://doi.org/10.6084/m9.figshare.30870761). These identified proteins were further categorized into six groups: 25 cell outer membrane proteins (OMPs, 32.9%), 9 periplasmic proteins (11.8%), 7 cell inner membrane proteins (IMPs, 9.2%), 13 extracellular proteins (17.1%), 8 cytoplasmic proteins (10.5%), and 14 other proteins (18.5%) (Fig. 2).

## Safety evaluation

The results demonstrated that no crucian carp deaths occurred during the 14-day observation period, and they remained in normal condition. Upon dissection, the liver, spleen, kidney, entire gut, heart, and other tissues appeared normal with no significant changes (Fig. 3A). Histopathological analysis using hematoxylin and eosin (H&E) staining revealed no significant pathological changes in the liver, spleen, kidney, midgut, and gill of crucian carp in both the experimental and PBS group (phosphate-buffered saline, PBS) (Fig. 3B through F).

## Serum antibody titer

The results of this study revealed a dynamic increase in serum antibody titers across all experimental groups (OMVs, Av, and OMVs + Av). At day 14, the OMV + Av group reached a titer of 1:320, which was significantly higher than that of the other three groups ($P <$ 0.05). At day 21, both the Av and OMV + Av groups attained a titer of 1:640, whereas the OMV group peaked at 1:320 by day 28. These results demonstrate that the Av and OMV + Av groups elicited significantly higher antibody titers compared to the OMV and PBS groups ($P < 0.05$), as summarized in Table 1. Overall, all three experimental groups enhanced serum antibody levels in crucian carp, with the most pronounced elevation observed in the OMV + Av group.

## Serum bactericidal activity

Statistical analysis of serum bactericidal activity using the drop plate counting method demonstrated that, although the PBS group displayed limited bactericidal capacity, the activity at a 1:1 dilution remained below 50% and was consequently designated as 0. These results are presented in Table 2. All experimental groups exhibited significantly elevated serum bactericidal activity compared to the PBS group, with activity levels inversely proportional to serum dilution. While serum bactericidal activities were not significantly different between the Av and OMV + Av groups ($P > 0.05$), the latter demonstrated superior bactericidal rates at the same dilution. Notably, at the 1:4 dilution, the OMV + Av group (61.21% ± 5.21%) exhibited outstanding and significant bactericidal activity compared with the other three groups ($P < 0.05$). All experimental groups demonstrated significant enhancement of serum bactericidal capacity in crucian carp compared to the PBS group, with the OMV + Av group achieving maximal bactericidal efficacy ($P < 0.05$). These findings suggest that OMVs in combination with Av potentiates the bactericidal activity of serum.

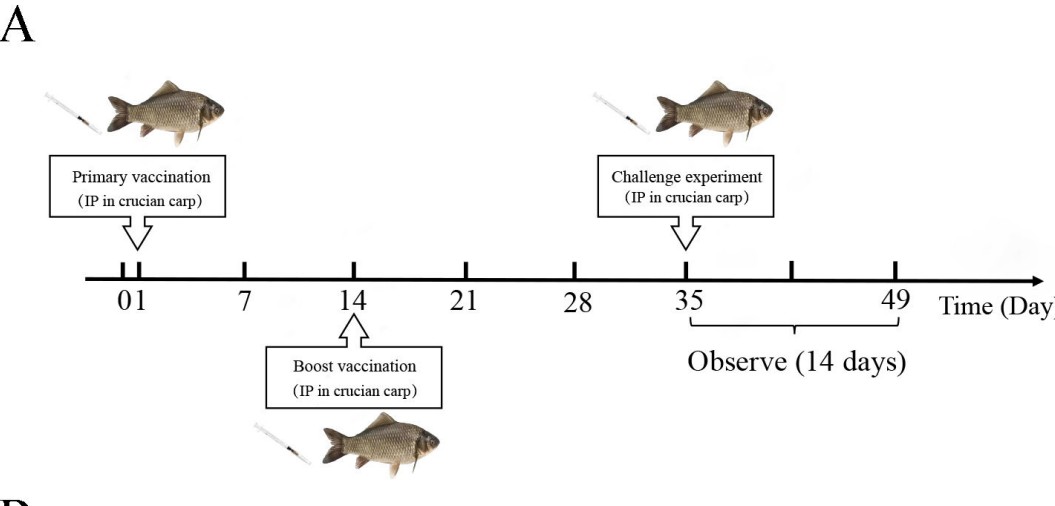

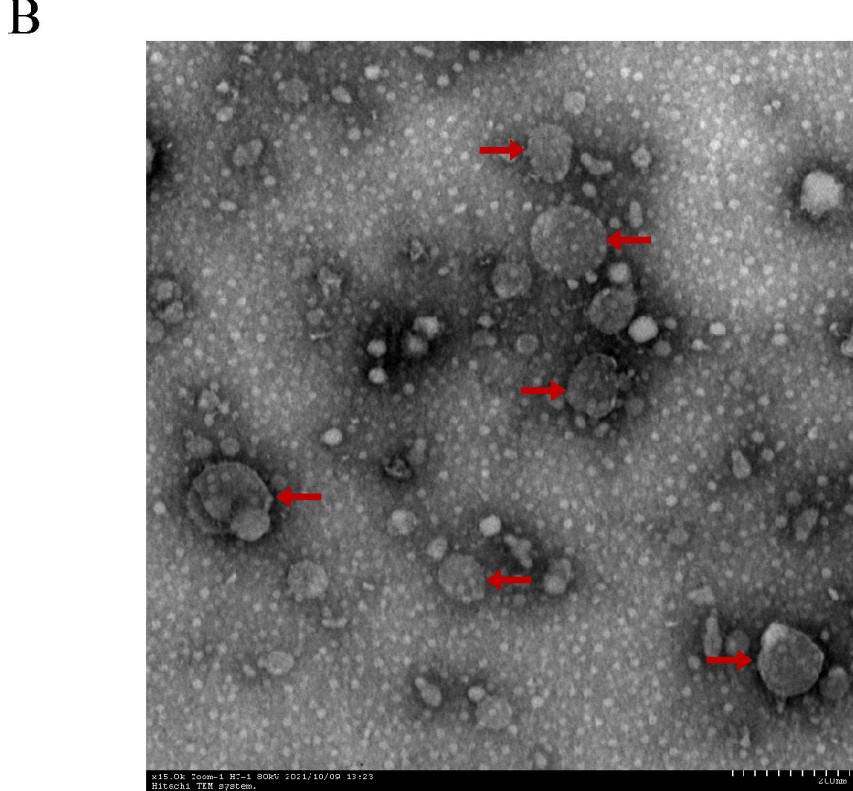

**FIG 1** Immune cycle and transmission electron micrographs. (A) Schematic diagram of the animal experimental immunization schedule; (B) transmission electron microscope results of *A. veronii* TH0426 OMVs.

## Serum-related immune parameter detection results

The levels of non-specific immune enzymes, including ACP, AKP, LZM, CAT, LDH, SOD, C3, and C4, as well as the overall level of IgM in the serum of crucian carp, are shown in Fig. 4. Compared to the PBS group, all experimental groups (OMV group, Av group, and OMV + Av group) exhibited significant immunoenhancing effects ($P < 0.05$). Specifically, the enhancing effects on ACP, LZM, CAT, and LDH were more pronounced in the Av group and the OMV + Av group than in the OMV alone group ($P < 0.05$). Notably, at day 35 post-immunization, the LZM content in the OMV + Av group was significantly higher than that in the Av group ($P < 0.05$). All experimental groups induced elevated levels of AKP, SOD, and complement C4. At day 21, the levels of AKP and C4 in both

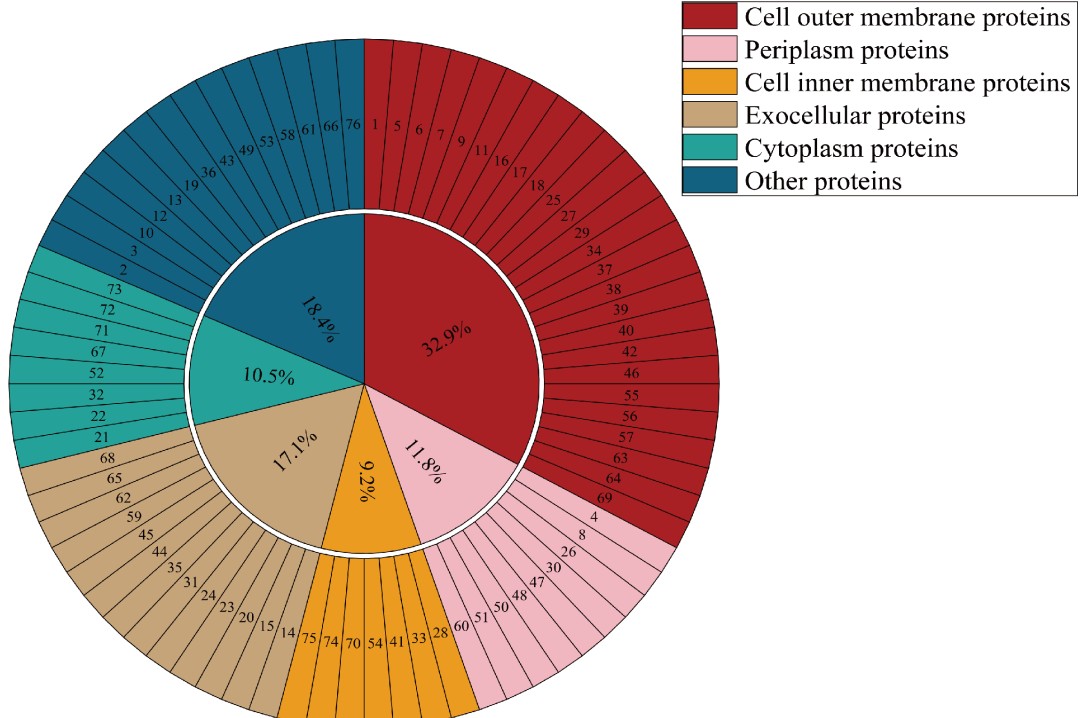

**FIG 2** Results of protein type analysis of OMVs from *A. veronii* TH0426. The first layer shows the proportions of six different categories of proteins, including cell outer membrane proteins (32.9%), periplasmic proteins (11.8%), cell inner membrane proteins (9.2%), extracellular proteins (17.1%), cytoplasmic proteins (10.5%), and other proteins (18.5%). The second layer presents the serial numbers of each protein, which can be referred to in Table S2 at https://doi.org/10.6084/m9.figshare.30870761.

the Av group and the OMV + Av group were significantly greater than those in the OMV group (*P* < 0.05). SOD levels showed an overall increasing trend across the groups. Complement C3 and IgM exhibited distinct response patterns: C3 levels were highest in the Av group, while IgM levels peaked in the OMV group. Importantly, at day 35, IgM levels in the OMV + Av group were significantly higher than those in the Av group (*P* < 0.05). Collectively, these results demonstrate that OMVs, either administered alone or in combination with Av, comprehensively enhance the immune defense capabilities of crucian carp. This enhancement was achieved by synergistically activating non-specific immune enzymes, the complement system, and specific antibody responses.

## The results of immune-related gene expression analysis in various tissues

The relative expression levels of genes encoding TGF-β, TNF-α, IL-10, and IL-1β in the liver, spleen, kidney, midgut, and gill of crucian carp are presented in Fig. 5. The relative expression levels of all four immune-related genes were upregulated to varying degrees across the tissues. *TGF-β* gene expression exhibited a sustained upregulation trend in all examined tissues. The OMV + Av group consistently demonstrated higher expression levels than the other three groups (OMV group, Av group, PBS group). Notably, at days 28 and 35 post-primary immunization, *TGF-β* expression levels in the spleen and gill of the OMV + Av group were significantly higher than those in both the Av group and the OMV group (*P* < 0.05). Furthermore, at day 35, midgut *TGF-β* expression in the OMV + Av group was significantly elevated compared to the Av group (*P* < 0.05). Overall, the expression levels of *TNF-α*, *IL-10*, and *IL-1β* genes were higher in both the Av group and the OMV + Av group compared to the OMV group and the PBS group (*P* < 0.05). Importantly, at day 28, *TNF-α* expression in the liver and IL-10 expression in the gill were both significantly higher in the OMV + Av group than in the Av group (*P* < 0.05). Collectively, these findings demonstrate that immunization significantly activated the

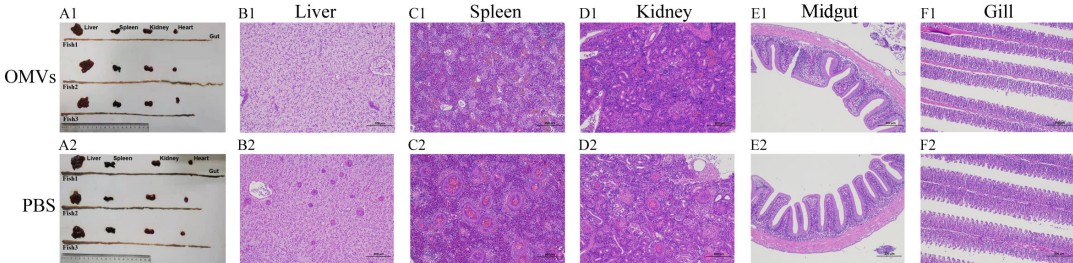

**FIG 3** Safety evaluation results. OMVs represent the experimental group, and PBS represents the control group. Panel A shows the anatomy of the crucian carp, while panels B–F show tissue sections of the crucian carp's liver, spleen, kidney, midgut, and gill.

expression of immunoregulatory and inflammation-related genes across the tissues of crucian carp. The combination of Av and OMVs showed superior efficacy in inducing *TGF-β* expression. Both the Av group and the OMV + Av group exhibited a stronger overall capacity to induce *TNF-α*, *IL-10*, and *IL-1β* expression compared to the OMV group and the PBS group.

## Analysis of immunoprotective effects

Challenge test analysis in Fig. 6 revealed a large quantity of mortality in the PBS group within 48 h post-challenge, culminating in 100% lethality within 7 days. Infected crucian carp displayed characteristic pathological manifestations, including hemorrhagic lesions and tissue ulceration. After 14 days of post-challenge monitoring, the OMV + Av group demonstrated a significantly higher relative percent survival than the OMV group ($P <$ 0.05). The OMV + Av group achieved the highest relative percent survival (66%), followed by the Av group (61%) and the OMV group (47%). These results indicate that all three experimental groups provided protective effects against infection by the *A. veronii* TH0426 strain.

## DISCUSSION

*A. veronii* is a significant pathogen threatening humans and fish, causing substantial economic losses in aquaculture (22, 23). In recent years, OMVs derived from pathogenic bacteria have garnered attention. These OMVs are enriched with various pathogen-associated molecular patterns (PAMPs) capable of activating the host's innate immune system, positioning them as a promising potential vaccine (24–27). Through systematic analysis of the biological characteristics and immunomodulatory functions of OMVs from the *A. veronii* TH0426 strain, this study further confirms the application potential as a vaccine.

OMVs were extracted and purified from the *A. veronii* TH0426 strain using ultrafiltration concentration (28). Analysis revealed the presence of small amounts of DNA. Studies suggest that secreted DNA can enhance OMV-host interactions, aiding in the evasion of host defense systems (29). The endotoxin concentration of the extracted OMVs in this study was 0.294 ng/mL (equivalent to 2.94 EU/mL). According to relevant provisions of the "Regulations on the Administration of Veterinary Drugs" (China), the standard endotoxin limit for veterinary vaccines is no higher than 50 EU/mL. This indicates that the

**TABLE 1** Changes in serum antibody titers of crucian carp[a]

| Group | Titer at immunization day: | | | | | |
|---|---|---|---|---|---|---|
| | 0 | 7 | 14 | 21 | 28 | 35 |
| PBS | 1:20[a] | 1:40[a] | 1:20[a] | 1:40[a] | 1:40[a] | 1:40[a] |
| OMVs | 1:40[a] | 1:80[b] | 1:160[b] | 1:160[a] | 1:320[b] | 1:320[b] |
| AV | 1:40[a] | 1:80[b] | 1:160[b] | 1:640[b] | 1:640[c] | 1:640[c] |
| OMVs + AV | 1:40[a] | 1:80[b] | 1:320[c] | 1:640[b] | 1:640[c] | 1:640[c] |

[a]Dynamic changes in serum antibody titers. Values with different superscript letters in the same column indicate significant differences ($P < 0.05$).

**TABLE 2** Bactericidal activity of crucian carp serum[a]

| Group | Bactericidal rate of different serum dilution multiples (%) | | | | | | Bactericidal titer |
|---|---|---|---|---|---|---|---|
| | 1:1 | 1:2 | 1:3 | 1:4 | 1:5 | 1:6 | |
| PBS | 21.94 ± 2.49[a] | 6.88 ± 0.48[a] | 0[a] | 0[a] | 0[a] | 0[a] | 0 |
| OMVs | 71.88 ± 4.53[b] | 62.94 ± 8.77[b] | 50.32 ± 9.43[b] | 31.36 ± 9.77[b] | 20.17 ± 7.04[b] | 4.60 ± 2.94[ab] | 3 |
| AV | 79.63 ± 2.36[b] | 70.05 ± 6.89[b] | 61.79 ± 7.95[bc] | 51.32 ± 6.64[b] | 29.57 ± 3.74[b] | 6.96 ± 7.68[ab] | 4 |
| OMVs + AV | 78.75 ± 5.55[b] | 71.49 ± 4.89[b] | 64.11 ± 9.24[c] | 61.21 ± 5.21[c] | 30.82 ± 2.97[b] | 7.32 ± 3.31[b] | 4 |

[a]Data are expressed as the mean ± standard deviation (SD) ($n = 3$). Values with different superscript letters in the same column indicate significant differences ($P < 0.05$).

OMVs from the *A. veronii* TH0426 strain have a relatively low endotoxin content, suggesting a certain degree of safety. Furthermore, the diameter of the extracted OMVs ranged from 10 to 300 nm, consistent with the size characteristics of OMVs from *Escherichia coli*, *Pseudomonas aeruginosa*, and *Vibrio cholerae* (30).

In the proteomic analysis of OMVs, a total of 76 proteins were identified in this study. Notably, key functional proteins identified included outer membrane porin protein II (OmpII; 80% coverage) and TonB-dependent transporters (TBDTs; 46.4% coverage), both of which are involved in OMV substance transport processes (31, 32). Additionally, outer membrane protein A (OmpA; 44% coverage) is a hallmark protein of Gram-negative bacterial OMVs and an important candidate target for subunit vaccine development (33). Collectively, these findings demonstrate that the OMVs extracted in this study are enriched in PAMPs, which may directly activate host immune pathways. This suggests that OMVs inherently possess the potential to serve as a vaccine antigen delivery system.

Immunogenicity evaluation demonstrated that OMVs administered alone or in combination with inactivated *A. veronii* vaccine significantly improved the serum antibody titers and bactericidal activity in crucian carp. This aligns with reports showing that OMVs from *Tenacibaculum maritimum* enhanced antibody levels in juvenile turbot (*Scophthalmus maximus* L.) (34), and OMVs from *Neisseria meningitidis* (serogroup X) boosted serum bactericidal activity in mice (35), confirming that OMVs effectively activate immune responses as a vaccine.

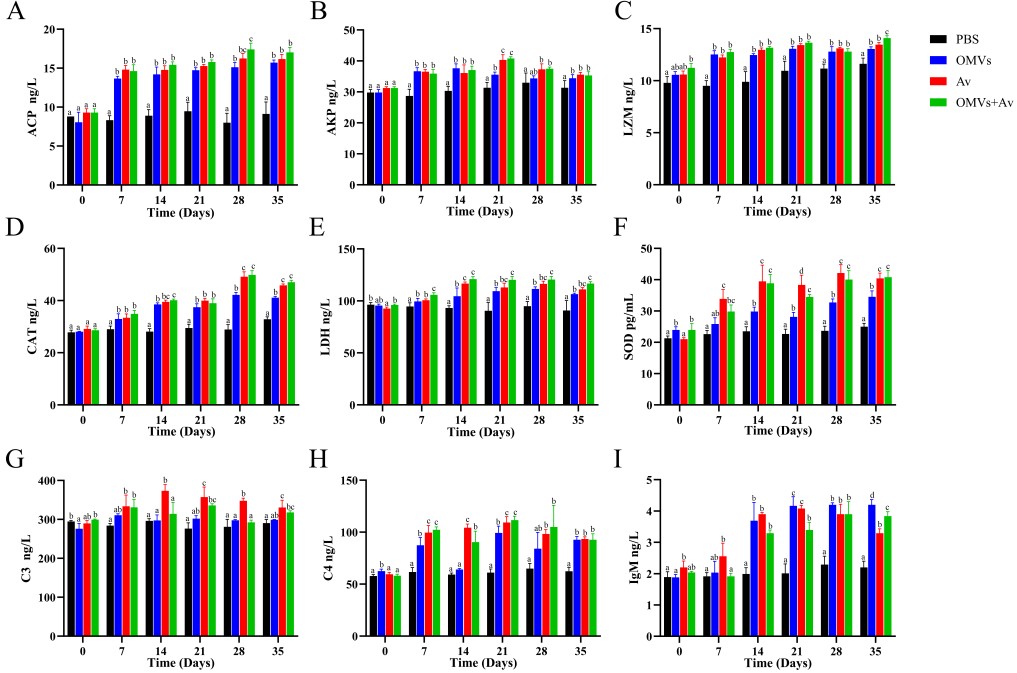

**FIG 4** Serum-related immune indicators detection and analysis results. (A) ACP; (B) AKP; (C) LZM; (D) CAT; (E) LDH; (F) SOD; (G) C3; (H) C4; (I) IgM. Data are presented as the mean ± standard deviation (SD) ($n = 3$); different letters indicate significant differences between groups ($P < 0.05$).

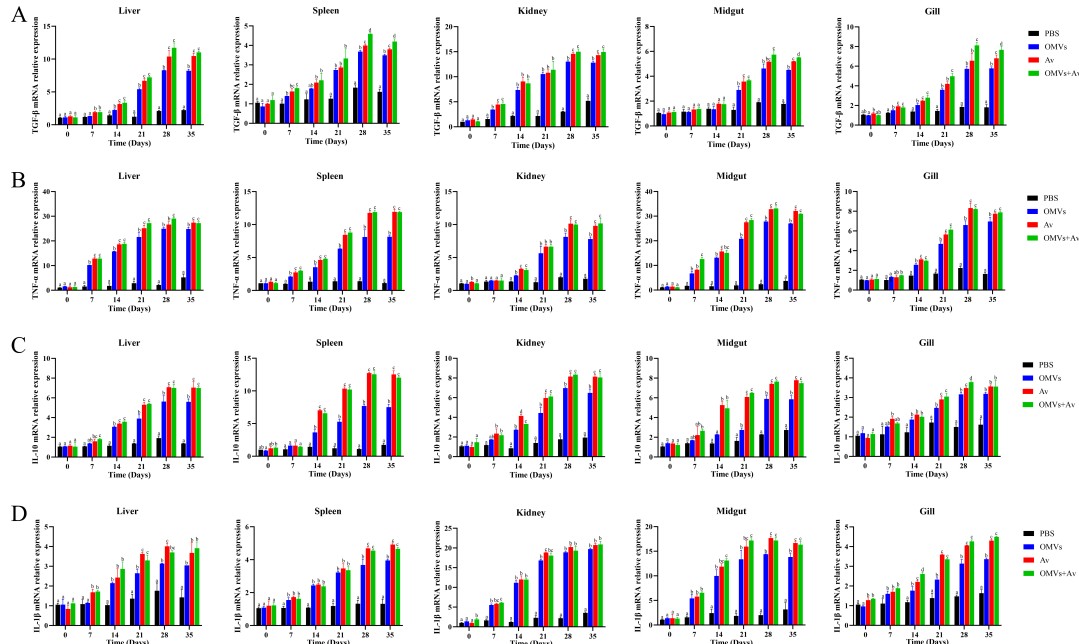

**FIG 5** Cytokine expression analysis. (A) TGF-β; (B) TNF-α; (C) IL-10; (D) IL-1β. Data are presented as the mean ± standard deviation (SD) ($n = 3$); different letters indicate significant differences between groups ($P < 0.05$).

Assessment of immune indicators revealed that *A. veronii* OMVs regulate crucian carp immune responses through multiple pathways. OMVs alone or combined with inactivated *A. veronii* vaccine significantly increased the levels of ACP, AKP, LZM, CAT, LDH, and SOD. Notably, the increases in ACP and LDH levels were more pronounced in the OMVs + Av group, suggesting a synergistic effect of OMVs and inactivated *A. veronii* vaccine that enhances defense against microbial infections and antioxidant stress capacity in crucian carp. This may be related to the activation of immune cell metabolism by PAMPs carried by OMVs. The induced increases in CAT and SOD levels effectively balanced reactive oxygen species (ROS) levels, while the elevated LZM levels strengthened bactericidal defense through enhanced lytic action (36–40). Serum levels of complement components C3 and C4, as well as IgM, were significantly higher in all three experimental groups compared to the PBS group. The Av group showed a distinct advantage in C3 levels, while the OMV group exhibited the highest IgM levels, followed by the OMV + Av group. This indicates that OMVs may upregulate complement component expression via Toll-like receptor (TLR) pathways and enhance specific antibody responses due to their nanoscale structure, which facilitates recognition by antigen-presenting cells (APCs) (34, 41–43). In summary, OMVs possess the dual function of activating non-specific immune enzyme systems and modulating humoral immune molecules. Their synergistic application with inactivated *A. veronii* vaccines further enhances the immune-boosting effect, providing experimental evidence for its potential as a vaccine.

Cytokines, as core mediators of the immunoregulatory network, participate in anti-infective immune processes by regulating cell development and immune cell responses (44). In this study, OMVs from *A. veronii*, administered alone or in combination with inactivated *A. veronii* vaccine, significantly upregulated the expression of TGF-β, TNF-α, IL-10, and IL-1β in various tissues of crucian carp. The upregulatory effect in the OMV + Av group was generally superior to that of either component alone, suggesting that OMVs may enhance immune responses through synergistic modulation of the cytokine network. TGF-β expression increased significantly after the secondary immunization, peaking at day 28. The sustained high expression in the OMV + Av group indicates its potential role in maintaining immune homeostasis by balancing pro-inflammatory and anti-inflammatory responses (45). The widespread upregulation

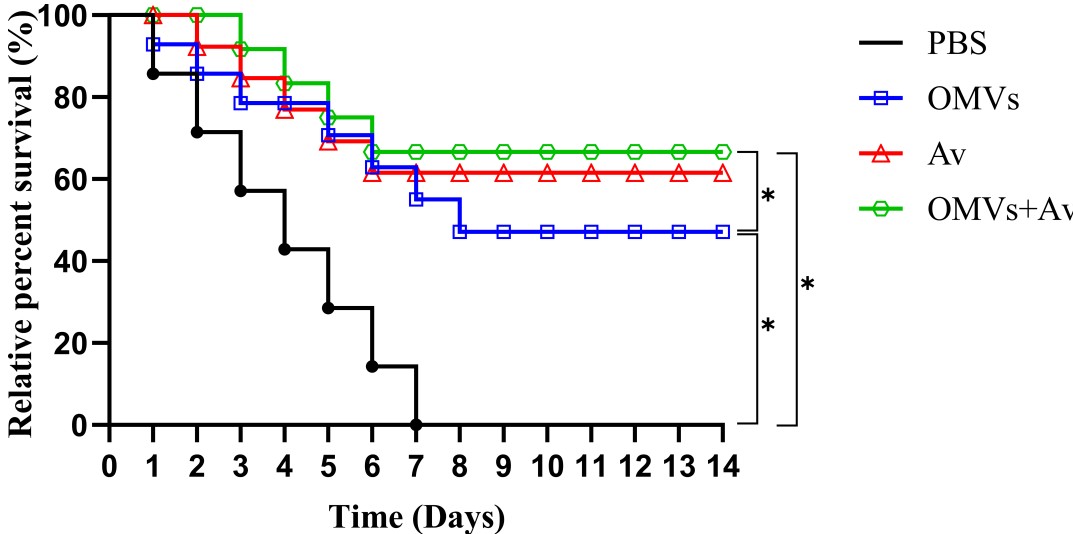

**FIG 6** Relative percent survival of crucian carp immunized with PBS, OMVs, Av, or OMVs + Av following challenge with the *A. veronii* TH0426 strain on day 35 post-immunization. Twenty fish/group were used to record the percent survival for 14 days. Differences at *P* < 0.05 (*) were considered significant between groups.

of the pro-inflammatory cytokines TNF-α and IL-1β confirms the potential of OMVs to activate innate immunity (46–48). In this study, the high expression of the *TNF-α* gene in various tissues of crucian carp is consistent with the report that OMVs from *Actinobacillus actinomycetemcomitans* can upregulate *TNF-α* in macrophages (46). The significant increase in IL-1β supports the notion that OMVs may enhance protection by activating inflammasome signaling pathways, consistent with the effect of *Escherichia coli* (BL21) OMVs inducing IL-1β secretion in mice (49). The peak expression of the anti-inflammatory cytokine IL-10 at day 28 and its predominant effect in the OMV + Av group suggest that it plays a role in maintaining immune balance by limiting excessive inflammatory damage (50, 51). In summary, OMVs can induce cellular immune responses while synergistically optimizing the activation efficiency of cytokine networks in combination with inactivated bacteria. This provides evidence for the immune mechanism of OMVs as a potential vaccine. The results of the challenge test showed that all experimental groups exhibited strong anti-infection ability. Among them, the OMV + Av combination group achieved a relative percent survival (RPS) of 66%, which could effectively resist pathogenic bacterial infection. Relevant studies have reported that Zhang et al. used an inactivated *A. veronii* vaccine to immunize largemouth bass, resulting in an RPS of 56.67% (52); Feng et al. reported that after immunizing zebrafish with the live attenuated vaccine of *Aeromonas dhakensis* quadruple gene deletion mutant Δ*aart*, the RPS reached 56.67% (53); Yao et al. developed an aerolysin mutant subunit vaccine (GS115-Ntaer), with a challenge survival rate of only 33.43% in the zebrafish model (54). The above comparisons indicate that the combined immunization strategy of OMVs and inactivated *A. veronii* bacteria adopted in this study has obvious advantages in inducing host anti-infection immunity and can significantly enhance the immune protection level of crucian carp against *A. veronii*.

The efficient immune response synergistically induced by OMVs and the inactivated *A. veronii* vaccine may better meet the dual requirements of commercial vaccines for safety and efficacy. However, this study acknowledges that the current preparation cost of OMVs is relatively high. Nevertheless, as a novel candidate vaccine, OMVs still hold potential value. Reducing the preparation cost of OMVs to enhance their practical application value will be one of the important directions for future research.

This study is in the exploratory validation phase of OMVs' vaccine potential, with the core objective to determine whether OMVs possess the ability to elicit an immune response in crucian carp and provide disease protection. Given that aquatic animals

(particularly fish) possess a more primitive immune system compared to mammals, natural antigens alone may elicit weak immune responses in certain aquaculture environments or animal strains, making it challenging to rapidly validate their vaccine value (55). Regarding CFA, its potent adjuvant effect effectively enhances the immune response to antigens in experimental animals, facilitating the observation and validation of target antigen immunogenicity in laboratory settings (56). Therefore, we employ CFA as an "immunostimulant" to ensure the experimental system can effectively detect immune response signals. Due to CFA's severe toxicity, it is unsuitable for practical vaccination (57). Therefore, future studies need to screen for suitable novel adjuvants to replace CFA.

## MATERIALS AND METHODS

### Bacterial strains, fish, and growth conditions

The *A. veronii* TH0426 strain (isolated from diseased *Pelteobagrus fulvidraco*; deposit numbers: MCCC 1K02718 and CCTCC AB2016061) was cultured in Luria-Bertani (LB) medium under shaking conditions (220 rpm) at 28°C. Healthy crucian carp (average weight, 50 ± 2.3 g; $n$ = 170) were obtained from an aquaculture facility in Changchun, China. The crucian carp were maintained in 200-L recirculating aquaria with controlled water parameters: temperature at 26 ± 1°C and dissolved oxygen at 6 mg/L. The crucian carp were fed twice daily with commercial extruded pellets (Tongwei, China) at 1% of total body weight. One-third of the water volume was exchanged every 48 h, with water quality maintained by biological filtration coupled with ozone sterilization. The health status of the crucian carp was assessed daily through visual inspection of morphological characteristics, locomotor activity, and feeding response (58). Following a 7-day acclimatization period under these conditions, the experimental procedures were initiated.

### Extraction and purification of OMVs

OMVs from *A. veronii* TH0426 were isolated and purified through an ultrafiltration-based concentration protocol (59), as follows: A 1:100 inoculum of *A. veronii* TH0426 was transferred to 1 L of LB broth and incubated at 28°C with orbital shaking at 220 rpm. After incubation for about 5 h (logarithmic growth phase, $OD_{600}$ = 0.6–1.1), the bacteria were filtered. Primary centrifugation (8,000 × $g$, 15 min, 4°C) was performed to collect the supernatant. The collected supernatant was then filtered through 0.22-μm pore-size membranes. The supernatant was filtered through 10-kDa molecular weight cutoff filters (3,000 × $g$, 30 min, 4°C), with the process repeated twice to achieve a 10-fold volume reduction. Ultracentrifugation (120,000 × $g$, 2 h, 4°C) was performed to pellet OMVs, followed by removal of the supernatant. The pellet was resuspended in 100 mL PBS. Secondary ultracentrifugation (120,000 × $g$, 2 h, 4°C) was performed for vesicle purification, retaining the final OMV-enriched pellet. The pellet contained purified OMVs, which were then resuspended in 2 mL PBS, aliquoted into four tubes of 0.5 mL each, and stored at −80°C until use.

### SDS-PAGE analysis of OMVs

To detect the molecular weight distribution of proteins within the OMVs, the OMVs were subjected to sodium dodecyl sulfate-polyacrylamide gel electrophoresis (SDS-PAGE) analysis according to the method described in reference (60). Gels were prepared with 12% separating and 5% stacking components using a commercial kit (Sangon Biotech, China). Protein samples (20 μL per lane) were loaded onto the gel. Electrophoresis was terminated when the bromophenol blue indicator migrated to the bottom of the gel. Gels were stained with Coomassie Brilliant Blue R-250 for 1.5 h at room temperature and then destained in solution until a clear background was achieved. Protein bands were visualized using a gel imaging system (Bio-Rad ChemiDoc, USA).

## DNA detection of OMVs

To detect the concentration and size distribution of DNA in OMVs, a micro-spectropho-tometer was used to determine the DNA concentration. Separately, a 50 µL aliquot of the OMV suspension was incubated at 37°C for 10 min to remove the excess RNA. After incubation, the samples were mixed with DNA loading buffer (5×) in a 1:4 ratio for electrophoretic analysis. *A. veronii* TH0426 genomic DNA (100 ng) was used as a positive control, and nuclease-free water was used as a blank control.

## Endotoxin detection of OMVs

Endotoxin levels in OMV suspensions were quantified using an ELISA kit (Coibo Bio, China) following the manufacturer's protocol. The plate was pretreated, and 100 µL of horseradish peroxidase (HRP)-conjugated detection antibody was dispensed into the standard and sample wells, excluding the blank controls. Plates were sealed with sealing film and incubated at 37°C for 60 min in a water bath. After discarding the liquid, the wells were filled with 1 mL of wash buffer and incubated for 1 min before removal. This washing cycle was repeated five times with fresh buffer. Substrate solutions A and B were added per well and incubated at 37°C for 15 min under light-protected conditions. Reactions were terminated with 50 µL of stop solution, and optical density was measured at 450 nm ($OD_{450}$) within 15 min using a microplate reader.

## BCA method for detecting the protein concentration

The protein concentration of OMVs was determined using a Bicinchoninic Acid (BCA) Protein Assay Kit (Beyotime Biotechnology, China) following the manufacturer's instructions. The working reagent was prepared by mixing Reagent A with Reagent B at a 1:50 ratio. Standard curve samples and test samples were dispensed in triplicate (25 µL/well) into a 96-well microplate. Following sample loading, 200 µL of working reagent was added to each well with brief vortex mixing (30 s), followed by incubation at 37°C for 30 min. After cooling to room temperature, the absorbance at 562 nm ($OD_{562}$) was measured using a microplate reader. The protein concentration of the samples was calculated based on the standard curve, the sample absorbance values, and the dilution factor.

## Detecting the morphology of OMVs

To analyze the morphology of OMVs by transmission electron microscopy (TEM), a 20 µL aliquot of purified OMVs, resuspended in PBS, was transferred onto a pre-cleaned glass slide using a micropipette. A copper grid was then positioned on the droplet of the vesicle suspension and allowed to float for sample adsorption, after which excess suspension was absorbed using filter paper. The grid was then negatively stained by immersion in a 2% phosphortungstic acid (PTA) solution droplet for 2 min. Residual stain was thoroughly blotted from the copper grid with filter paper, followed by air-drying. The prepared specimen was ultimately examined using a transmission electron microscope (TEM) (61).

## Mass spectrometry analysis of OMVs

The purified OMVs were submitted to Shanghai Bioprofile Technology Company (China) for proteomic characterization. The measurements were performed using liquid chromatography-tandem mass spectrometry (LC-MS/MS) (62). Specifically, the proteins were enzymatically digested to generate peptides, which were then desalted with C18 StageTip columns and subsequently vacuum-dried. The dried peptides were subsequently reconstituted in 0.1% formic acid (FA), and their concentration was determined by measuring the absorbance at 280 nm ($OD_{280}$). The protein composition was then analyzed using LC-MS/MS.

## Safety evaluation of animal experiments

Ten crucian carp, which had been acclimated for 1 week, were selected to carry out the safety evaluation experiment (58) and were randomly divided into two groups, with five crucian carp in each group. The purified OMV suspension was adjusted to a concentration of 600 µg/mL using PBS solution. The first group was set as the control group and was intraperitoneally (IP) injected with PBS solution (0.1 mL per fish). The second group was intraperitoneally injected with OMVs (600 µg/mL, 0.1 mL per fish). The health status and mortality of the crucian carp were observed and recorded over a 2-week period. After 2 weeks, three crucian carp from each group were randomly selected, anesthetized, and dissected to collect tissues such as the liver, spleen, kidney, midgut, and gill. These tissues were then processed for paraffin embedding and sectioning. The histological changes in the main tissues of the test group and the PBS group were observed and analyzed.

## Vaccine preparation

The vaccine group protocol was adapted from established methodology (58). The OMVs were diluted to a concentration of 300 µg/mL using PBS and then emulsified by mixing them in a 1:1 ratio with complete Freund's adjuvant (CFA) (OMV group). Subsequently, the *A. veronii* TH0426 bacterial suspension was adjusted to a concentration of $1 \times 10^8$ CFU/mL using PBS, and 1% formaldehyde solution was added. The suspension was kept at 4°C for 24 h. Afterward, it was centrifuged at 8,000 × $g$ for 10 min to collect the inactivated bacteria. The supernatant was discarded, and the bacterial pellet was resuspended in an equal volume of PBS. The suspension was centrifuged again at 8,000 × $g$ for 10 min to wash the bacteria, and the washing process was repeated three times to remove the formaldehyde. After confirming sterility by plating on LB agar plates, the bacteria were mixed with an equal volume of complete CFA to prepare the inactivated bacterial preparation of *A. veronii* TH0426 (inactivated *A. veronii* vaccine [Av] group). The OMVs (300 µg/mL) were then mixed in equal volumes with the inactivated *A. veronii* TH0426 ($1 \times 10^8$ CFU/mL) to create a mixture of OMVs and inactivated bacteria (OMV + Av group).

## Animal immunization

Animal immunization was conducted according to relevant references (58, 63). A total of 160 crucian carp were randomly divided into four groups, with 40 crucian carp per group. The first group, the PBS group, served as the control group, where each crucian carp was injected intraperitoneally with 0.2 mL of PBS solution. The second group, designated as the OMV group, was administered an intraperitoneal injection of 0.2 mL of OMVs (150 µg/mL). The third group, designated as the Av group, was administered an intraperitoneal injection of 0.2 mL of inactivated *A. veronii* TH0426 ($5 \times 10^7$ CFU per fish). The fourth group was the OMV + Av group, where each crucian carp was injected intraperitoneally with 0.1 mL of OMVs (300 µg/mL) and 0.1 mL of inactivated *A. veronii* TH0426 ($1 \times 10^8$ CFU/mL) mixed in equal volumes. The primary immunization was administered on day 1 of the experiment, followed by a booster immunization on day 14. The immunization schedule is detailed in Fig. 1A.

## Sample collection

Blood and tissue samples were collected on days 0, 7, 14, 21, 28, and 35 during the rearing period. At each sampling point, three healthy crucian carp were randomly selected from each group. All crucian carp were fasted for 24 h prior to sampling (64, 65). The crucian carp were then anesthetized, and blood was collected from the caudal vein. Approximately 1 mL of whole blood per crucian carp was transferred into a standard 1.5 mL Eppendorf tube. Serum was separated by centrifugation at 2,000 × $g$ for 15 min. Serum samples were stored at −80°C for subsequent analysis of serum antibody titers, serum bactericidal activity, and enzyme levels. Samples of liver, spleen, kidney, gut, and

gill tissues were collected and immediately snap-frozen in liquid nitrogen. The tissues were stored at −80°C for subsequent immune gene expression analysis.

## Serum antibody titer detection

The serum antibody titer of crucian carp was detected using established protocols (66). Bacterial suspensions ($2 \times 10^8$ CFU/mL) were heat-inactivated at 60°C for 30 min in a thermostatic water bath. Using PBS as the control group, different proportions of the serum to be tested were added to a 96-well plate with a micropipette. Then, 50 µL of the heat-inactivated bacterial suspension was added to each well, and the mixture was uniformly mixed by pipetting. The plate was incubated at 37°C for 1 h, followed by overnight storage at 4°C. The agglutination results were observed, the experiment was repeated three times, and the serum antibody titer was analyzed with reference to relevant literature (67, 68).

## Serum bactericidal activity assay

The serum bactericidal activity assay was performed based on reference (69). *A. veronii* TH0426 bacterial suspension was diluted in PBS to $1 \times 10^3$ CFU/mL. Serum samples were collected and heat-inactivated at 56°C for 40 min in a thermostatic water bath, followed by dilution with PBS to generate twofold serial dilutions (1:1 to 1:6). Using PBS as the control group, 5 µL of *A. veronii* TH0426 suspension ($1 \times 10^3$ CFU/mL) was mixed with either 50 µL of untreated serum or heat-treated serum. The mixture was incubated at 30°C for 65 min. Each dilution was spot-plated for viable bacterial enumeration. The bactericidal rate (%) was calculated as follows: (1 − the number of cells surviving in the serum-treated bacterial suspension/the number of cells surviving in the PBS-treated bacterial suspension) × 100%. Results were reported using the serum dilution factor at which the bactericidal rate was greater than 50%. The experiment was repeated three times, and the evaluation criteria referred to relevant literature (70).

## Serum enzyme-linked immunosorbent assay

The enzyme-linked immunosorbent assay (ELISA) method was used to detect the levels of acid phosphatase (ACP), alkaline phosphatase (AKP), lysozyme (LZM), catalase (CAT), lactate dehydrogenase (LDH), superoxide dismutase (SOD), complements (C3, C4), and the overall level of immunoglobulin M (IgM) in the serum of crucian carp (71). The experimental procedure was performed in strict accordance with the instructions provided by the ELISA kits (Coibo Bio, China) and was repeated three times. Research data were analyzed by referencing relevant literature (72).

## Cytokine expression analysis

To assess the immunomodulatory effects of *A. veronii* TH0426-derived OMVs, real-time quantitative PCR (RT-qPCR) was employed to analyze the expression of immune-related genes (transforming growth factor-beta [*TGF-β*], tumor necrosis factor-alpha [*TNF-α*], interleukin-10 [*IL-10*], and interleukin-1beta [*IL-1β*]) in the liver, spleen, kidney, midgut, and gill (73). The research data were analyzed by referring to relevant literature (72). The reverse transcription was performed according to the instructions of the PrimeScript RT Reagent Kit (Takara, Japan), and the cDNA samples were stored at −80°C after reverse transcription. The 20 µL reaction mixture contained: 10 µL of TB Green Premix Ex Taq II (2×), 0.8 µL of forward primers (10 µM), 0.8 µL of reverse primer (10 µM), 0.4 µL of ROX Reference Dye II (50×), 2 µL of cDNA template, and 6 µL of sterile water. Three replicates were performed for each sample, with sterile water as the negative control and β-actin as the internal reference gene (74). The primer sequences are shown in Table S1 at https://doi.org/10.6084/m9.figshare.30870761. The gene expression levels were calculated using the $2^{-\triangle\triangle CT}$ method (75).

## Challenge test

A challenge protection test was conducted at 35 days post-primary immunization to evaluate vaccine efficacy (76). Twenty crucian carp from each group were randomly selected and intraperitoneally injected with 0.2 mL of *A. veronii* TH0426 suspension ($2 \times 10^7$ CFU/mL). The health status and mortality of the crucian carp were monitored daily for 14 consecutive days. The number of deaths in each group was recorded. The relative percent survival (RPS) was calculated as follows: RPS = [1 − (mortality of the immunized group/mortality of the control group)] × 100%.

## Data analysis

All statistical analyses were performed using SPSS v26.0 (IBM, USA) and GraphPad PRISM v8.4.0 (GraphPad Software, USA). One-way analysis of variance (ANOVA) was used to compare differences between groups. Differences were analyzed using Duncan's multiple range test, with a significance level established at $P < 0.05$.

## ACKNOWLEDGMENTS

This work was supported by the Shandong Natural Science Foundation Youth Fund Project (ZR20230D024) and Guangdong Basic and Applied Basic Research Foundation (2025A1515012527).

Conceptualization: Z.-X.W. and D.-J.A. Methodology: Z.-X.W., D.-J.A., and Y.-X.K. Formal analysis: D.-J.A. and Y.-X.K. Investigation: Q.Y., Z.-Q.Z., and L.C. Resources: G.-P.G. and G.-S.G. Writing-original draft: Z.-X.W. Writing-review and editing: B.-T.Y. and Y.-H.K. Visualization: Z.-X.W., D.J.A., and Y.-X.K. Project administration: B.-T.Y. and Y.-H.K.

## AUTHOR AFFILIATIONS

[1]Marine College, Shandong University, Weihai, China
[2]Shenzhen Research Institute of Shandong University, Shandong University, Shenzhen, China
[3]Institute of Animal and Veterinary Sciences, Jilin Academy of Agricultural Sciences, Changchun, China
[4]College of Veterinary Medicine/College of Animal Science and Technology, Jilin Agricultural University, Changchun, China
[5]Hebei Key Laboratory of Preventive Veterinary Medicine, Hebei Normal University of Science & Technology, Qinhuangdao, China

## AUTHOR ORCIDs

Bin-Tong Yang http://orcid.org/0000-0003-1709-2282
Yuan-Huan Kang http://orcid.org/0000-0002-1379-1126

## FUNDING

| Funder | Grant(s) | Author(s) |
| --- | --- | --- |
| Natural Science Foundation of Shandong Province | ZR2023QD024 | Yuan-Huan Kang |
| Basic and Applied Basic Research Foundation of Guangdong Province | 2025A1515012527 | Yuan-Huan Kang |

## AUTHOR CONTRIBUTIONS

Zong-Xiu Wu, Conceptualization, Methodology, Visualization, Writing – original draft | Ding-Jie An, Conceptualization, Formal analysis, Methodology, Visualization | Yi-Xuan Kou, Formal analysis, Methodology, Visualization | Qing Yang, Investigation | Zhi-Qiang Zhang, Investigation | Li Chen, Investigation | Guang-Ping Gao, Resources | Gui-Sheng

Gao, Resources | Bin-Tong Yang, Project administration, Writing – review and editing | Yuan-Huan Kang, Project administration, Writing – review and editing

## DATA AVAILABILITY

All data generated or analyzed during this study are included in this article and its supplemental material, which are available at https://doi.org/10.6084/m9.figshare.30870761.

## ETHICS APPROVAL

This study was conducted under the approval of the Institutional Animal Care and Use Committee of Jilin Agricultural University. All animal experimental procedures were performed in strict accordance with the Jilin Agricultural University Animal Experiment Regulations (JLAU08201409) and the National Institutes of Health Guide for the Care and Use of Laboratory Animals (NIH Publication No. 1438023).

## ADDITIONAL FILES

The following material is available online.

Open Peer Review

**PEER REVIEW HISTORY (review-history.pdf).** An accounting of the reviewer comments and feedback.

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
