## [Reviewer comments · Microbiology Spectrum]

Microbiology Spectrum

Outer membrane vesicles from *Aeromonas veronii*: Biological properties and synergistic immunity enhancement with inactivated bacteria in crucian carp

Zongxiu Wu, Dingjie An, Yixuan Kou, Qing Yang, Zhiqiang Zhang, Li Chen, Guangping Gao, Guisheng Gao, Bintong Yang, and Yuanhuan Kang

Corresponding Author(s): Yuanhuan Kang, 山东大学

Review Timeline:

Submission Date:	August 12, 2025
Editorial Decision:	September 11, 2025
Revision Received:	November 10, 2025
Accepted:	November 27, 2025

Editor: Monica Cartelle Gestal

Reviewer(s): Disclosure of reviewer identity is with reference to reviewer comments included in decision letter(s). The following individuals involved in review of your submission have agreed to reveal their identity: wenbin wang (Reviewer #1); Chentao Lin (Reviewer #2); Erdal Özbek (Reviewer #3)

Transaction Report:

DOI: <https://doi.org/10.1128/spectrum.02497-25>

Re: Spectrum02497-25 (Outer membrane vesicles from *Aeromonas veronii*: Biological properties and synergistic immunity enhancement with inactivated bacteria in crucian carp)

Dear Dr. Yuanhuan Kang:

Thank you for the privilege of reviewing your work. Below you will find my comments, instructions from the Spectrum editorial office, and the reviewer comments.

Revision Guidelines

Sincerely,
Monica Cartelle Gestal
Editor
Microbiology Spectrum

Reviewer #1 (Comments for the Author):

The manuscript of Wu and coworkers reported the biological properties and synergistic immunity enhancement of OMV from *Aeromonas veronii* when used together with inactivated bacteria in crucian carp. The experiment design and the writing and presentation of this manuscript is clear. Nevertheless, several questions and suggestions were listed below for this article before publication.

1. Line 57, What's the meaning of self-adjuvant vaccine platform? But the results obtained in this study is based on the adoption of CFA adjuvant

Results

2. Line 126, colorless is very normal under TEM, consider revision

3. Line 154, control group, please make it clear (PBS)

4. Line 143-200, specific values and the general criteria to assess them are needed in the results to better compression.

Discussion

5. Line 258-260, This suggests that OMVs inherently possess the potential to serve as a vaccine antigen delivery system and may not require the addition of exogenous adjuvants. How to exclude the effect of CFA on all the indicators since all the experiment group used the CFA adjuvant

6. Line 320, the survival of the OMVs+ Av group was 66%. What are the criteria for evaluating aquatic animal vaccines? The protection rates of the OMV group and the whole bacteria group are also very close, so what's the advantage of OMVs+ Av group considering the high costs. Is there any difference in cross-protection of different *A. veronii* strains, *Aeromonas* strains and the protection period?

7. The comparison of the protection rate with the previous studies for *Aeromonas* is also encouraged in the discussion part.

Methods part

8. Line 369-370, What is the proportion of free proteins in OMVs? Have you measured the particle size distribution?

9. Line 387, What is the amount of genomic DNA used in each well? What is the sensitivity of electrophoresis detection of DNA? What is the concentration of DNA in OMVs?

10. Line 390, The commonly used method for endotoxin detection is the LAL method. How accurate is this antibody method for endotoxin detection? Has the endotoxin of the extracted *A. veronii* been verified?

Supporting information

11. Fig.S1A, the names of each lane are not listed in the caption

12. Table S2, the A/B/C in the column header should be replaced with the concrete content

Reviewer #2 (Comments for the Author):

The manuscript by Wu et al. characterizes outer membrane vesicles (OMVs) from *Aeromonas veronii* TH0426 and evaluates their immunogenicity and vaccine potential in crucian carp. The study is generally well-designed, the data presented is comprehensive and supports the main conclusions, and the writing is clear. However, several points require clarification and additional mechanistic depth to strengthen the manuscript significantly.

1. The manuscript repeatedly attributes the immune effects to the "self-adjuvant" property of OMVs, likely mediated via TLR pathways (e.g., Lines 285-287, 315-317). However, this remains speculative based on the current data. In this manuscript the authors prepared vaccine CFA are used in OMVs group (OMVs with CFA Lines 447-449) and inactivated *A. veronii* vaccine (Av) group (lines 457-459), but without a group only use OMV or CFA alone as control. The absence of a group immunized with inactivated bacteria adjuvanted with CFA makes it difficult to benchmark the efficacy of the OMVs as an adjuvant. Is the effect of OMVs+Av superior to, or merely comparable to, the gold-standard adjuvant CFA? The authors also discuss this question in line 336-338. To add significant mechanistic depth, the authors could consider: An in vitro experiment using carp macrophages or other immune cells, demonstrating that OMV stimulation leads to the upregulation of TLR genes (e.g., TLR2, TLR4). The authors should focus on demonstrates the potential of *A. veronii* OMVs as a new vaccine but not as adjuvant.

2. Line 32-34, OMV Proteomic analysis identified 76 proteins, but the data are shown in coverage (OmpII; 80% coverage, OmpA; 44% coverage), and Table S2. The abundance of 76 proteins can represent the relative concentration in the OMV, which is more importance.

3. In Fig. 1B, Under transmission electron microscopy the size of the OMVs varied, with diameters ranging from approximately 10 to 300 nm. Usually OMV particle sizes are analysis by Nanoparticle Tracking Analysis (NTA).

4. Regarding serum antibody titers (Table 1), the text states (Lines 144-146) that the serum analysis revealed significantly elevated antibody titers in immunized crucian carp, peaking at day 28. However, the AV group and OMVs+Av group reach the peak at 1:640 (days 21), the OMVs group reach the peak at 1:320 (days 28).The wording should be adjusted to accurately reflect the data, perhaps highlighting the boost (Day 14).

10 November, 2025

Monica Cartelle Gestal

Editor

Microbiology Spectrum

Dear Monica Cartelle Gestal and Reviewers,

Re: Revised Manuscript Spectrum02497-25

On behalf of all co-authors, I would like to thank you and the two reviewers very much for your favorable comments and constructive suggestions on our manuscript (MS) (**Manuscript No: Spectrum02497-25**). We have carefully studied and responded to the reviewers' comments and suggestions. For clarity, we used the “tracked changes” mode in the WORD to show the revised/changed text and sentences in the revised MS. Two MS files are uploaded: one shows a “clean file” as “manuscript”, and the other is a “tracked changes” as the “Marked-up manuscript”. In the following, we detail our point-by-point responses to these specific comments and suggestions.

Responses to the comments and suggestions of Reviewer #1:

General comments:

The manuscript reported the biological properties and synergistic immunity enhancement of OMV from *Aeromonas veronii* when used together with inactivated bacteria in crucian carp. The experiment design and the writing and presentation of this manuscript are clear. Nevertheless, several questions and suggestions were listed below for this article before publication.

Responses: Thank you for your valuable comments and advice, which have been valuable and helpful while revising and improving our manuscript. We have carefully considered the suggestions. We have also checked and revised the entire manuscript thoroughly as suggested. Please find below our point-by-point responses. We hope that the revision we have made meets your expectations.

Point 1: Line 57, What's the meaning of self-adjuvant vaccine platform? But the results obtained in this study is based on the adoption of CFA adjuvant

Response: Thank you for your question. According to literature reports,

bacteria-derived OMVs possess a self-adjuvant effect (the dual functions of efficiently delivering antigens and activating immune responses independently without the addition of exogenous adjuvants (1-4)). Therefore, we included content related to the adjuvant effect in the introduction. However, focusing on the present study, we did not demonstrate the adjuvant function of OMVs; the purpose of this study is to explore the application potential of OMVs as a vaccine. Consequently, the expressions regarding the adjuvant effect in the manuscript were inappropriate, and we have revised the relevant content accordingly.

1. Zha TM, Cai YL, Jiang YJ, He XM, Wei YQ, Yu YF, Tian XH. 2023. Vaccine adjuvants: mechanisms and platforms. *Signal Transduct Target Ther*, 8:283.
2. Lyu JH, Liou GG, Wang M, Kan MC. 2025. The genetic, biophysical and immunological studies of a self-adjuvanted protein nanoparticle. *Vaccine*, 56:127087.
3. Ahmed AAQ, Besio R, Xiao L, Forlino A. 2023. Outer membrane vesicles (OMVs) as biomedical tools and their relevance as immune-modulating agents against *H. pylori* infections: Current status and future prospects. *Int J Mol Sci*, 24:8542.
4. Li M, Zhou H, Yang C, Wu Y, Zhou XC, Liu H, Wang YC. 2020. Bacterial outer membrane vesicles as a platform for biomedical applications: An update. *J Control Release*, 323:253–268.

Point 2: Line 126, colorless is very normal under TEM, consider revision

Response: Thank you for your professional suggestions. The inappropriate expression here is indeed an oversight in our manuscript writing process. We have revised the relevant content in the paper accordingly and sincerely thank you again for your guidance.

Point 3: Line 154, control group, please make it clear (PBS)

Response: Thank you for pointing out this issue. We have now clearly specified the control group in the manuscript.

Point 4: Line 143-200, specific values and the general criteria to assess them are needed in the results to better compression.

Response: Thank you for your professional suggestions. In accordance with your comments, we have supplemented and improved the relevant content in the manuscript. Thank you again for your guidance.

Point 5: Line 258-260, This suggests that OMVs inherently possess the potential to serve as a vaccine antigen delivery system and may not require the addition of exogenous adjuvants. How to exclude the effect of CFA on all the indicators since all the experiment group used the CFA adjuvant.

Response: Thank you for raising this question. The original purpose of this experiment was to evaluate the application potential of OMVs as a vaccine. During the experiment, we found that OMVs do possess antigen delivery capabilities, which is consistent with the findings of relevant literature, indicating their potential as a vaccine antigen delivery system. However, the adjuvant effect of OMVs cannot be directly confirmed in this experiment, so the relevant expressions were inappropriate. Thank you for your correction, we have revised and improved the relevant content accordingly.

Point 6: Line 320, the survival of the OMVs+ Av group was 66%. What are the criteria for evaluating aquatic animal vaccines? The protection rates of the OMV group and the whole bacteria group are also very close, so what's the advantage of OMVs+ Av group considering the high costs. Is there any difference in cross-protection of different *A. veronii* strains, *Aeromonas* strains and the protection period?

Response: Thank you for your professional questions. We address each of your inquiries as follows:

Currently, there is no unified and strict evaluation standard for the efficacy of aquatic animal vaccines. Based on the analysis of relevant literature reports (1, 2), the results of this study show that the survival rates of the OMVs+Av group (66%), Av group (61%), and OMVs group (47%) all reached moderate or higher protection levels, among which the OMVs+Av group exhibited the optimal protective efficacy.

Regarding the advantages of the OMVs+Av group: The core purpose of this experiment was to verify the application potential of OMVs as a vaccine, so it did not involve considerations of OMV production processes and practical application costs. Based on existing technologies, the production cost of OMVs is indeed relatively high.

From the perspective of subsequent vaccine industrialization, OMVs do not temporarily have a cost advantage. This issue is expected to be addressed through future technological innovations to reduce production costs and enhance their practical application value.

Both cross-protection and vaccine protection period are important issues worthy of in-depth exploration, but this study only serves as a preliminary evaluation of the role of OMVs: this study selected the *Aeromonas veronii* TH0426 strain as the research object, and did not conduct in-depth research on the evaluation of cross-protective effects against other strains. Meanwhile, only the protective rate was initially determined, and no systematic study was carried out on the vaccine protection period of OMVs. The above two issues will be the focus of subsequent research. Thank you again for your professional suggestions, which have provided important guidance for the improvement of our research and the direction of future work.

1. Jung MH, Kole S, Jung SJ. 2022. Efficacy of saponin-based inactivated rock bream iridovirus (RBIV) vaccine in rock bream (*Oplegnathus fasciatus*). Fish Shellfish Immunol 121:12-22.
2. Giovanni A, Shi YZ, Wang PC, Tsai MA, Chen SC. 2025. Comparative evaluation of oral biofilm and killed cell vaccines against *Streptococcus iniae* in Four-finger threadfin fish (*Eleutheronema tetradactylum*): Immune response and protection efficacy. J Fish Dis, 0:e70062.

Point 7: The comparison of the protection rate with the previous studies for *Aeromonas* is also encouraged in the discussion part.

Response: Thank you for your professional guidance and suggestions. We have added a comparative analysis of the protection rate from this study versus previous studies on *Aeromonas* to the discussion section.

Point 8: Line 369-370, What is the proportion of free proteins in OMVs? Have you measured the particle size distribution?

Response: Thank you for your question. In this study, OMVs were extracted and purified using ultrafiltration concentration combined with ultracentrifugation, a method that can effectively separate OMVs from free proteins. Since the core objective of this study was to obtain high-purity OMVs, the specific proportion of free

proteins was not accurately determined. Regarding vesicle size, preliminary analysis was performed using transmission electron microscopy (TEM) (1, 2), and the results showed that the particle size range was approximately 10-300 nm. No further in-depth analysis of OMVs' particle size has been conducted yet.

1. Liu GN, Ma NN, Cheng KM, Feng QQ, Ma XT, Yue YL, Li Y, Zhang TJ, Gao XY, Liang J, Zhang LZ, Wang XW, Ren ZH, Fu YX, Zhao X, Nie GJ. 2024. Bacteria-derived nanovesicles enhance tumour vaccination by trained immunity. *Nat Nanotechnol* 19:387-398.
2. Augustyniak D, Olszak T, Drulis-Kawa Z. 2022. Outer membrane vesicles (OMVs) of *Pseudomonas aeruginosa* provide passive resistance but not sensitization to LPS-specific phages. *Viruses* 14:121.

Point 9: Line 387, What is the amount of genomic DNA used in each well? What is the sensitivity of electrophoresis detection of DNA? What is the concentration of DNA in OMVs?

Response: Thank you for your professional question. To ensure clear electrophoresis bands without overloading, the loading amount of genomic DNA per well in this study was strictly controlled at 100 ng, and the relevant experimental details have been fully supplemented in the paper. Since this study only required a preliminary determination of whether DNA is present in OMVs, agarose gel electrophoresis—an operationally convenient method suitable for preliminary screening (with relatively low sensitivity)—was adopted. In addition, we have measured the DNA concentration of the preserved OMV samples, and the result was 25.34 ng/μL. This data has also been added to the manuscript. Thank you again for your careful guidance.

Point 10: Line 390, The commonly used method for endotoxin detection is the LAL method. How accurate is this antibody method for endotoxin detection? Has the endotoxin of the extracted *A. veronii* been verified?

Response: Thank you for your professional question. Referring to relevant literature (1, 2), this study employed the enzyme-linked immunosorbent assay (ELISA) for endotoxin detection and did not adopt the limulus amoebocyte lysate (LAL) assay for this purpose. Since the core objective of this study is to explore the vaccine potential of OMVs, with a primary focus on their immunogenicity and protective efficacy, no

further in-depth analysis of endotoxin in the vesicles was conducted. In future studies, we will further optimize and improve the relevant methods. Thank you again for your valuable suggestions.

1. Violi F, Cammisotto V, Bartimoccia S, Pignatelli P, Carnevale R, Nocella C. 2023. Gut-derived low-grade endotoxaemia, atherothrombosis and cardiovascular disease. *Nat Rev Cardiol* 20:24-37.
2. Schneier M, Razdan S, Miller AM, Briceno ME, Barua S. 2020. Current technologies to endotoxin detection and removal for biopharmaceutical purification. *Biotechnol Bioeng* 117:2588-2609.

Point 11: Fig.S1A, the names of each lane are not listed in the caption

Response: Thank you for your question. We have added the corresponding names for each lane in Figure S1A.

Point 12: Table S2, the A/B/C in the column header should be replaced with the concrete content

Response: Thank you for your inquiry. We have replaced the “A/B/C” in the column headers of Table S2 with specific content.

Responses to the comments and suggestions of Reviewer #2:

General comments:

The manuscript characterizes outer membrane vesicles (OMVs) from *Aeromonas veronii* TH0426 and evaluates their immunogenicity and vaccine potential in crucian carp. The study is generally well-designed, the data presented is comprehensive and supports the main conclusions, and the writing is clear. However, several points require clarification and additional mechanistic depth to strengthen the manuscript significantly.

Responses: Thank you for your valuable comments and suggestions. We have improved and modified the content of the article by referring to the suggestions provided by the reviewer.

Point 1: The manuscript repeatedly attributes the immune effects to the "self-adjuvant" property of OMVs, likely mediated via TLR pathways (e.g., Lines 285-287, 315-317). However, this remains speculative based on the current data. In this manuscript the authors prepared vaccine CFA are used in OMVs group (OMVs with CFA Lines 447-449) and inactivated *A. veronii* vaccine (Av) group (lines 457-459), but without a group only use OMV or CFA alone as control. The absence of a group immunized with inactivated bacteria adjuvanted with CFA makes it difficult to benchmark the efficacy of the OMVs as an adjuvant. Is the effect of OMVs+Av superior to, or merely comparable to, the gold-standard adjuvant CFA? The authors also discuss this question in line 336-338. To add significant mechanistic depth, the authors could consider: An in vitro experiment using carp macrophages or other immune cells, demonstrating that OMV stimulation leads to the upregulation of TLR genes (e.g., TLR2, TLR4). The authors should focus on demonstrates the potential of *A. veronii* OMVs as a new vaccine but not as adjuvant.

Response: Thank you for your professional guidance. The original purpose of this study was to explore the application potential of OMVs as a vaccine, so CFA adjuvant was used in all experimental groups. Based on the existing research results, the adjuvant effect of OMVs cannot be directly confirmed, for which we have revised the relevant expressions in the manuscript. In subsequent research, we will consider conducting in vitro experiments on crucian carp immune cells for further investigation. Thank you again for your valuable suggestions.

Point 2: Line 32-34, OMV Proteomic analysis identified 76 proteins, but the data are shown in coverage (OmpII; 80% coverage, OmpA; 44% coverage), and Table S2. The abundance of 76 proteins can represent the relative concentration in the OMV, which is more importance.

Response: Thank you for your question. The purpose of using mass spectrometry in this study was to preliminarily explore the types of proteins contained in OMVs; therefore, in-depth analysis of protein abundance was not performed. Protein abundance is indeed of great significance for result analysis. Your suggestion has reminded us that more comprehensive considerations are needed in subsequent experiments. Thank you for your guidance.

Point 3: In Fig. 1B, Under transmission electron microscopy the size of the OMVs varied, with diameters ranging from approximately 10 to 300 nm. Usually OMV particle sizes are analysis by Nanoparticle Tracking Analysis (NTA).

Response: Thank you for your professional question. The preliminary purpose of this study was to understand the particle size of OMVs. Therefore, referring to relevant literature (1, 2), we observed and determined the approximate diameter range of OMVs by TEM, and have not yet adopted the NTA technology you mentioned for in-depth research. Your suggestions have provided professional guidance for our follow-up research. Thank you for your professional guidance.

1. Liu GN, Ma NN, Cheng KM, Feng QQ, Ma XT, Yue YL, Li Y, Zhang TJ, Gao XY, Liang J, Zhang LZ, Wang XW, Ren ZH, Fu YX, Zhao X, Nie GJ. 2024. Bacteria-derived nanovesicles enhance tumour vaccination by trained immunity. *Nat Nanotechnol* 19:387-398.
2. Augustyniak D, Olszak T, Drulis-Kawa Z. 2022. Outer membrane vesicles (OMVs) of *Pseudomonas aeruginosa* provide passive resistance but not sensitization to LPS-specific phages. *Viruses* 14:121.

Point 4: Regarding serum antibody titers (Table 1), the text states (Lines 144-146) that the serum analysis revealed significantly elevated antibody titers in immunized crucian carp, peaking at day 28. However, the AV group and OMVs+Av group reach the peak at 1:640 (days 21), the OMVs group reach the peak at 1:320 (days 28). The

wording should be adjusted to accurately reflect the data, perhaps highlighting the boost (Day 14).

Response: Thank you for your professional guidance and valuable suggestions. The issue you pointed out is indeed an oversight in the process of writing our manuscript, and we have carefully revised the relevant content in the paper accordingly. We sincerely appreciate your meticulous review and corrections again.

Re: Spectrum02497-25R1 (Outer membrane vesicles from *Aeromonas veronii*: Biological properties and synergistic immunity enhancement with inactivated bacteria in crucian carp)

Dear Dr. Yuanhuan Kang:

Your manuscript has been accepted, and I am forwarding it to the ASM production staff for publication. Your paper will first be checked to make sure all elements meet the technical requirements. ASM staff will contact you if anything needs to be revised before copyediting and production can begin. Otherwise, you will be notified when your proofs are ready to be viewed.

Sincerely,
Monica Cartelle Gestal
Editor
Microbiology Spectrum

Reviewer #1 (Comments for the Author):

The questions for this article have been well addressed.

Reviewer #2 (Comments for the Author):

The authors have addressed my concerns.